# The wide spectrum anti-inflammatory activity of andrographolide in comparison to NSAIDs: A promising therapeutic compound against the cytokine storm

**Mitchell Low**[1]*, **Harsha Suresh**[1,2], **Xian Zhou**[1], **Deep Jyoti Bhuyan**[1], **Muhammad A. Alsherbiny**[3], **Cheang Khoo**[4], **Gerald Münch**[2], **Chun Guang Li**[1]

1 NICM Health Research Institute, Western Sydney University, Penrith, Australia, 2 School of Medicine, Western Sydney University, Campbelltown, Australia, 3 Faculty of Pharmacy, Pharmacognosy Department, Cairo University, Cairo, Egypt, 4 Wentworth Institute of Higher Education, Surry Hills, Sydney, Australia

* Mitchell.low@westernsydney.edu.au

**Data Availability Statement:** All relevant data are within the paper and its Supporting Information files.

## Abstract

The challenges of the COVID-19 pandemic have highlighted an increasing clinical demand for safe and effective treatment options against an overzealous immune defence response, also known as the "cytokine storm". Andrographolide is a naturally derived bioactive compound with promising anti-inflammatory activity in many clinical studies. However, its cytokine-inhibiting activity, in direct comparison to commonly used nonsteroidal anti-inflammatory drugs (NSAIDs), has not been extensively investigated in existing literature. The anti-inflammatory activities of andrographolide and common NSAIDs, such as diclofenac, aspirin, paracetamol and ibuprofen were measured on lipopolysaccharide (LPS) and interferon-γ induced RAW264.7 cells. The levels of PGE2, nitric oxide (NO), TNF-α & LPS-induced release of pro-inflammatory cytokines on differentiated human macrophage THP-1 cells were measured against increasing concentrations of andrographolide and aforementioned NSAIDs. The associated mechanistic pathway was examined on NFκB using flow cytometry on the human endothelial-leukocyte adhesion molecule (ELAM9) (E-selectin) transfected RAW264.7 cells with green fluorescent protein (GFP). Andrographolide exhibited broad and potent anti-inflammatory and cytokine-inhibiting activity in both cell lines by inhibiting the release of IL-6, TNF-α and IFN-γ, which are known to play a key role in the etiology of cytokine storm and the pathogenesis of inflammation. In comparison, the tested NSAIDs demonstrated weak or no activity against proinflammatory mediators except for PGE2, where the activity of andrographolide ($IC_{50}$ = 8.8 μM, 95% CI = 7.4 to 10.4 μM) was comparable to that of paracetamol ($IC_{50}$ = 7.73 μM, 95% CI = 6.14 to 9.73 μM). The anti-inflammatory action of andrographolide was associated with its potent downregulation of NFκB. The wide-spectrum anti-inflammatory activity of andrographolide demonstrates its therapeutic potential against cytokine storms as an alternative to NSAIDs.

**Funding:** This study was partially supported by a Research Partnership Grant (RPG) from the Western Sydney University and LIPA Pharmaceuticals (Prof. Nikolaus Sucher), and a Western Sydney University Research Grant Scheme (RGS) grant (Prof. Nikolaus J. Sucher and Prof. Gerald Münch).

**Competing interests:** NO authors have competing interests.

## Introduction

In light of the recent coronavirus disease SARS-CoV-2 (COVID-19) pandemic, the development of immunomodulatory drugs has gained considerable interest from the public and the scientific community. This interest has emerged due to the identification of "hyperinflammatory" acute respiratory distress syndrome (ARDS) as the key driver of the severity and mortality of COVID-19, which is supported by findings from early and recent clinical trials. [1–4]. In a range of auto-immune and infectious conditions, immune-signalling proteins known as cytokines are released from the host immune system, which can go into overdrive and trigger the uncontrolled surging levels of cytokine release, the "cytokine storm" or cytokine release syndrome (CRS) [5]. The characteristic "cytokine storm" of COVID-19 is also a primary feature in patients who experience sudden acute respiratory syndrome (SARS) and middle-east respiratory syndrome (MERS), which are caused by other coronaviruses [6]. Clinically, it commonly presents as systemic inflammation, multiple organ failure, and high inflammatory parameters that can lead to patient mortality [7]. The recent (2023) uptick in cases of respiratory syncytial virus (RSV) in Australia [8,9], especially among children, also has cytokine storm associated with it [10] and is accompanied by even more dangerous secondary complications such as encephalitis [11]. Hence, it is imperative to contemplate the utilisation of anti-inflammatory therapies and immunosuppressive drugs that can target a broad range of inflammatory mediators to prevent a fatal cytokine storm and hence adverse patient outcome. However, more rigorous comparative studies, *in vitro* followed by *in vivo*, are needed to confirm the activity and efficacy of any novel and off-label drugs [12].

Nonsteroidal anti-inflammatory drugs (NSAIDs) are the regular prescription for acute and chronic inflammation-induced pain [13,14]. As a result, they are also being explored for managing fever and hyperinflammation that can occur during viral infections. NSAIDs work by inhibiting cyclooxygenase-1 (COX-1) and cyclooxygenase-2 (COX-2), thereby blocking the production of prostaglandins, which are essential mediators of fever and inflammation. Ibuprofen, an NSAID frequently prescribed, has demonstrated the ability to decrease interleukin-6 (IL-6) levels in human tissues and sputum [15]. This observation aligns with outcomes from clinical trials conducted amid the COVID-19 pandemic. These trials and more recent meta-analyses suggest that utilising ibuprofen and other NSAIDs for anti-IL-6 therapies could be a viable option, since NSAIDs do not cause increased rates of SARS-COV-2 infection or symptom severity when used for analgesic and antipyretic treatment during COVID-19 [16,17]. However, caution is advised when using ibuprofen and other NSAIDs to treat severe COVID-19 that requires hospitalisation [14,18]. Serious adverse effects on the gastrointestinal (GI) and cardiovascular systems may limit the use of NSAIDs against cytokine storm. For instance, diclofenac showed an increased risk of myocardial infarction, similar to rofecoxib with compatible high COX-2 inhibitory potency [19], suggesting a link. NSAIDs may also increase angiotensin-converting enzyme 2 expression, increase the viral load, and may worsen the clinical outcomes [20]. Thus, due to the associated risks and adverse reactions, the use of NSAIDs such as ibuprofen and acetaminophen for cytokine storm remains controversial [21].

Natural products and their derivatives have long formed "the backbone of modern pharmacopoeias" [22,23]. There is an increasing realisation that alternatives to synthetic approaches to drug development are needed, thus renewing interest in natural products as drug discovery sources [24]. Andrographolide is an ent-Labdane diterpenoid, and the primary bioactive compound from *Andrographis paniculata*, a medicinal herb, has been used to treat a wide variety of ailments linked to inflammation [25–27]. Andrographolide has been shown to inhibit a wide range of inflammatory mediators and can be a therapeutic candidate for a wide range of inflammatory and bacterial conditions, including rheumatoid arthritis, acute colitis, cigarette

smoke-induced oxidative lung injury, *Chlamydia trachomatis* infections, and in some instances of bacterial pneumonia [28–31]. Notably, andrographolide inhibited influenza A virus-induced inflammation in the C57BL/6 mice model by reducing key cytokines of the cytokine storm, including IL-6, IL-10, TNF-α and interferon (IFN)-γ via the downregulation on NFκB and JAK-STAT signaling pathway [32].

The safety profile of andrographolide has been well established in literature and is considered to quite safe [33]. Adverse events associated with andrographolide, and andrographolide-derivative medications are extremely rare, but include gastrointestinal problems, skin and subcutaneous disorders, and anaphylaxis. These adverse events are primarily associated with injections, oral consumption of andrographolide and andrographolide herbal extracts are essentially safe [34]. Therefore, andrographolide is a potential lead compound for the development of new anti-inflammatory compounds guided by historical use and not by specific COX inhibition. Interestingly, andrographolide has been reported to exhibit gastro-protective, and ulcer-preventive effects, which, combined with its well-documented anti-inflammatory effects, could make it a safe alternative to traditional NSAIDs [35].

This study aimed to investigate the effect of andrographolide on lipopolysaccharide (LPS) and IFN-γ induced inflammatory mediators and cytokines on macrophage and monocyte cells. The inhibitory effects of andrographolide in suppressing a range of cytokines were compared with popular NSAIDs such as aspirin, ibuprofen, diclofenac, and acetaminophen.

## Materials and methods

### Chemicals and reagents

Andrographolide ($C_{20}H_{30}O_5$, purity >98%) was purchased from Biopurify Phytochemicals Ltd. (Chengdu, China), with reported purity certified by HPLC analysis. Ibuprofen sodium salt (98%), diclofenac (99%), acetaminophen (paracetamol, 99%), prednisone (98%) and dexamethasone (97%) analytical standards were purchased from Sigma-Aldrich (NSW, Australia).

Lipopolysaccharides (LPS) isolated from *Escherichia coli* strain 0111:B4, phorbol 12-myristate 13-acetate (PMA), 3-(4,5-dimethylthiazol-2-yl)-2,5-diphenyltetrazolium bromide (MTT) powder, 3,3,5,5- tetramethylbenzidine (TMB), dimethylsulfoxide (DMSO), and citric acid analytical standards were purchased from Sigma-Aldrich (NSW, Australia). The murine IFN-γ and murine TNF-α, and prostaglandin E2 (PGE2) enzyme-linked immunosorbent assay (ELISA) kit were purchased from Peprotech (NSW, Australia). Dulbecco's Modified Eagle Medium (DMEM) and Roswell Park Memorial Institute (RPMI) used to culture cells were obtained from Lonza (NSW, Australia). GlutaMax, penicillin, and streptomycin were purchased from Life Technologies (NSW, Australia). The foetal bovine serum (FBS) (French origin) was purchased from Bovogen Biologicals (VIC, Australia). The strep avidin horse radish peroxidase used in the TNF-α ELISA was purchased from BD Biosciences (NSW, Australia). The Bio-Plex Pro cytokine, chemokine and growth factor assay kits, human cytokine 17 and 27-plex, were purchased from Bio-Rad (NSW, Australia).

### Cell culture

The murine macrophage RAW264.7 cells (from American Type Culture Collection (ATCC), VA, USA) were maintained in Dulbecco's Modified Eagle Medium (DMEM) from Lonza (NSW, Australia), containing 4.5 g/L D-glucose and supplemented with 2 mM l-GlutaMax, 100 units/mL penicillin, 100 μg/mL streptomycin (Life Technologies, Australia), and 5% FBS. The immortalized RAW264.7 monocyte or macrophage-like cells originate from the Abelson leukemia virus transformed cell line derived from BALB/c mice [36]. The immortalized RAW264.7 cell line was preferred over human/animal derived primary cell lines due to a

cheaper-costs, ease of availability, and minimal ethical concerns [37]. The cells were incubated in a humidified atmosphere containing 5% $CO_2$ and 95% air. The human monocyte THP-1 cells were cultured in RPMI 1640 media from Lonza (NSW, Australia), containing 4.5 g/L D-glucose and supplemented with 2 mM GlutaMax, 100 units/mL penicillin, 100 μg/mL strepto-mycin, and 10% FBS from Life Technologies (NSW, Australia), at 37°C, 5% $CO_2$ in 95% air. Using PMA (100 nM) for 24 h, THP-1 cells were differentiated towards macrophage-like phe-notype and subsequently seeded for the bioassays.

## Protocol for MTT viability determination

A 100 μL of MTT solution (0.2 mg/mL MTT in complete medium) was added to the cell cul-ture and incubated for 2 h at 37°C (5% $CO_2$). The MTT solution was removed, and 150 μL of DMSO was added to dissolve the formazan crystals. It should be noted DMSO was used as the vehicle control, and no noticeable effect in the final analysis. It should also be noted that dexamethasone (97%) was used as the cytokine positive control. The plate was shaken for 5 min before absorbance was measured at 595 nm on a FLUOstar Omega microplate reader from BMG Labtech (VIC, Australia). All relevant MTT assay data, standard curves, and normalizations are presented in the **S1 File** which contains MS Excel and GraphPad Prism files.

## Determination of nitric oxide release in LPS and IFN-γ stimulated RAW264.7 cells

Nitric oxide (NO) release was quantified using Griess reagent [38]. Briefly, the RAW264.7 cells were seeded at $1 \times 10^5$ cells/well on a 96-well culture plate (Corning® Costar®, Sigma-Aldrich, Australia) for 48 h. Andrographolide and the NSAIDs were dissolved in DMSO (final concentration of 0.1% *w/v*), 1 h before stimulation with LPS and IFN-γ (50 ng/mL, 50 units/mL). After the cells were co-incubated for 18 h, the supernatant was collected (180 μL) and reacted with the Griess reagent (100 μL) to quantify dissolved nitrates at 540 nm (colorimetry) on the FLUOstar microplate reader. The remaining cells were tested with MTT solution to assess the cell viability.

## Determination of prostaglandin E2 and TNF-α in LPS and IFN-γ stimulated RAW264.7 cells

Prostaglandin E2 (PGE2) release and TNF-α were quantified by commercial ELISA kits in accordance with supplying manufacturers' protocol [39]. Briefly, the RAW264.7 cells were seeded at $1 \times 10^5$ cells/well in a 96-well plate for 48 h. The compounds of interest were dissolved in DMSO (final concentration 0.1% *w/v*) 1 h before the stimulation with LPS and IFN-γ (50 ng/mL, 50 units/mL). After 18 h, the supernatant (180 μL) was collected and subjected to ELISA assay.

## Determination of multiple cytokines in LPS-stimulated THP-1 cells using a Bioplex cytokine assay

The inhibitory effect on cytokines was tested in a panel of 17 inflammatory mediators using Bio-Plex Pro cytokine, chemokine and growth factors assay kits, human cytokine 17 and 27-plex from Bio-Rad, (NSW, Australia) on PMA-differentiated THP-1 cells. The 17 inflam-matory mediators include proinflammatory cytokines such as TNF-α, IFN-γ, IL-1β, IL-5, IL-6, IL-7, IL-8, IL-12, G-CSF, GM-CSF, MCP-1 and MIP-1b, and anti-inflammatory cytokines such as IL-2, IL-4, IL-9, IL-10 and IL-13. Briefly, the cells were seeded at $1 \times 10^5$ cells/well in a

96-well plate for 48 h in PMA (100 nM). Non-adherent cells were removed by washing with fresh medium (PMA-free) and confluence observed at near 100%, with no significant cytotoxicity observed. Note that THP-1 cells are non-adherent, when THP-1 cells differentiate, they become adherent, therefore any undifferentiated THP-1 cells are washed away during this step. The compounds of interest were dissolved in DMSO (final concentration 0.1% *w/v*) for 1 h before the stimulation with LPS (1μg/mL) for 6 h before the supernatant was harvested and stored (-80°C) for analysis using the bead-based assay. The experiments were conducted with the Bio-Plex 100 system from Bio-Rad (NSW, Australia). The plates were washed using a 96-well plate magnetic handheld washer from Bio-Rad (NSW, Australia). The Bio-Plex Manager 3.0 software was used to operate the system and interpret the data.

## The regulation of NF-kB signalling pathway using a FACS Canto II flow cytometer

The RAW264.7 cells were stably transfected with the human endothelial-leukocyte adhesion molecule (ELAM9) (E-selectin) promoter (-760 to +60 mV), driving destabilised enhanced green fluorescent protein (GFP) [40,41]. The ELAM9 RAW264.7 cells used in the NF-$\kappa$B activation assay were analysed using a BD FACS Canto II flow cytometer from Becton, Dickinson and Company (NSW, Australia), equipped with a high throughput fluorescence-activated cell sorting (FACS) flow, with a medium flow rate, and an autosampler with three laser sets (405, 488, 635 nm) and corresponding filter sets. After the co-incubation with compounds of interest and LPS (50 μg/mL) and IFN-γ (50 units/mL) for 5.5 h after stimulation, the RAW264.7 cells were then washed with ice-cold (0°C) PBS and harvested by trypsin. After resuspension in PBS with 10% FBS, the cells were then filtered through a 50 μm Nylon filter into a new 96-well plate. The blue layer was used for the analysis of GFP in the ELAM9 RAW264.7 cells. The forward scatter (FSC) and side scatter (SSC) were determined based on the size and shape of the control cells plated in each experiment. Readout of the laser intensity was normalised in each experiment to the fluorescence of the normal RAW264.7 cells. The data was analysed using FlowJo v10 from BD Biosciences (NJ, USA). The response was then normalised to the unstimulated and stimulated controls and expressed as a percentage of stimulated untreated NF-kB activation. The dose-response curves were constructed in GraphPad Prism v5 (CA, USA) by plotting the log of the dose concentration against the percentage release. A non-linear 4-parameter variable slope dose-response curve was fitted to calculate the IC$_{50}$ value for each sample tested. All relevant flow cytometry data, including plots for normal and activated cells, cell counts, and flow rates are included in the S2 File. The flow cytometry files are listed in the FCS format, and can be accessed via the FlowJo v10 analysis software.

## Statistical analysis

All data is reported with the standard error of the mean (SEM) displayed as error bars in the figures. The *in vitro* experiments were performed in triplicate, and the entire experiment was repeated on three or more separate days (*n* = 3). Multicytokine assays were performed only once and in duplicate due to their high cost, and the assay duplication (*n* = 2) is indicated in the dose-response curves. The dose-response data was fitted with a log (inhibitor) *vs.* normalised response with a variable slope model using GraphPad Prism v5 (CA, USA). The IC$_{50}$ values were calculated from the fitted curves. All fitted curves were constrained to a minimum of 0 and a maximum of 100. The uncertain measurement in the IC$_{50}$ values is expressed at the 95% confidence interval (CI) (*p* = 0.05).

**Table 1. The half maximal inhibitory concentrations of andrographolide, diclofenac, aspirin, paracetamol, and ibuprofen in LPS and IFN-γ induced PGE2, NO and TNF-α expressions in murine RAW264.7 cells.**

| Compounds* | PGE2 | | NO | | TNF-α | |
|---|---|---|---|---|---|---|
| | $IC_{50}$ (μM) | 95% CI (μM) | $IC_{50}$ (μM) | 95% CI (μM) | $IC_{50}$ (μM) | 95% CI (μM) |
| Andrographolide | 8.80 | 7.4 to 10.4 | 7.4 | 6.7 to 8.1 | 23.3 | 20.1 to 27.0 |
| Diclofenac | ~ 0.01 | 0.001 to 0.1 | 222 | 169 to 292 | >333 | - |
| Aspirin | 14.10 | 10.1 to 19.7 | >1600 | - | >1600 | - |
| Paracetamol | 7.73 | 6.14 to 9.73 | 2763 | 2406 to 3174 | >6000 | - |
| Ibuprofen | 0.09 | 0.08 to 0.11 | 1058 | 949 to 1180 | 839 | 381 to 1845 |

* All cell viabilities were verified using the MTT assay and were found to be non-toxic at all tested concentrations. The data presented is the result of 3 independent experiments (n = 3).

## Results and discussion

### Broad inhibitory effects of andrographolide on NO, PGE2 and TNF-α in LPS and IFN-γ stimulated RAW264.7 cells

Andrographolide and common NSAIDs, including diclofenac, aspirin, paracetamol, ibuprofen were tested on LPS and IFN-γ induced $PGE_2$, NO and TNF-α assays on RAW264.7 cells at multiple concentration levels. The $IC_{50}$ values for all tests are summarised in **Table 1**.

The results from **Fig 1** indicate that andrographolide exhibited a dose-dependent inhibitory effect against $PGE_2$ ($IC_{50}$ = 8.8 μM, 95% CI = 7.4 to 10.4 μM), with activity comparable to paracetamol ($IC_{50}$ = 7.73 μM, 95% CI = 6.14 to 9.73 μM). All tested NSAIDs demonstrated greater potency than andrographolide, except for aspirin, with an $IC_{50}$ value of 14.10 μM (**Fig 1A**). Therefore, andrographolide inhibited $PGE_2$ production at about the same potency as weak non-selective NSAIDs such as aspirin and paracetamol *in vitro*. Diclofenac and ibuprofen demonstrated more potent $PGE_2$ inhibition, but they are not considered selective COX-2 inhibitors and long-term use is associated with adverse gastrointestinal (GI) effects [26], whereas paracetamol and aspirin are generally considered safer in the clinical context [42]. NSAIDs relieve pain and fever by inhibiting the synthesis of $PGE_2$ *via* COX enzymes.

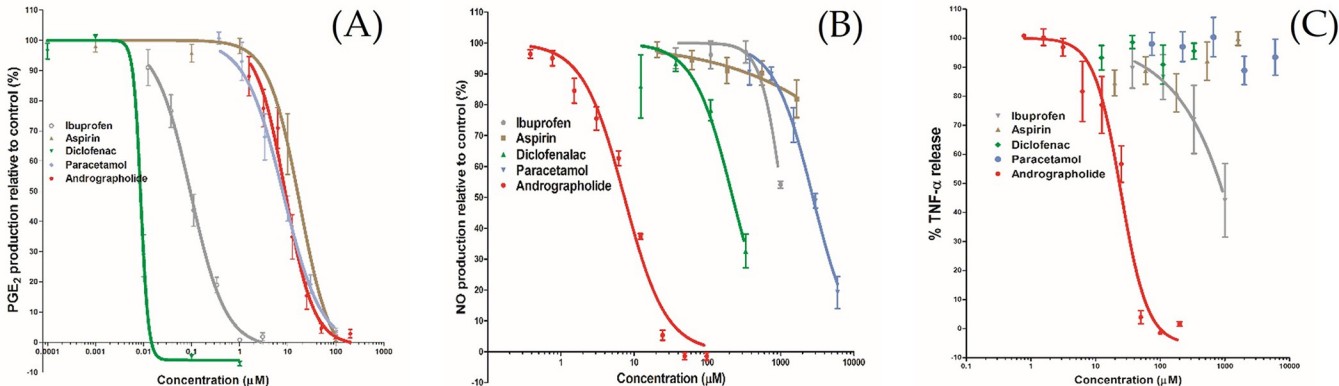

**Fig 1.** Dose-response curves of andrographolide compared with ibuprofen, aspirin, diclofenac, and paracetamol in inhibiting LPS and IFN-γ induced $PGE_2$ (A), NO (B) and TNF-α (C) expressions on RAW264.7 cells. The data has been fitted using a log (inhibitor) vs. normalised response curve with variable slope model (*n* = 3). The error bar expresses the standard error of the mean.

The inhibition of COX-2 leading to decreased proinflammatory cytokine levels and leukocyte activation is considered therapeutically beneficial, whereas the inhibition of COX-1 is associated with unwanted side effects partly attributed to the non-specific suppression of prostaglandins [35]. Since COX-1 has been associated with improved survival in viral-infected hyperinflammatory conditions, NSAIDs with non-specific COX inhibition are not recommended for therapeutic use in cytokine release syndrome (CRS) as the first line of clinical treatment [43]. In contrast, andrographolide has been shown to specifically inhibit COX-2 expression in human fibroblast cells under the stimulation of LPS [44], and andrographolide sodium bisulfate is known to exert a gastroprotective effect against indomethacin-induced gastric ulcer in rats *via* an increase of mRNA expression of COX-1 [45]. Thus, the *in vitro* anti-PGE2 activity of andrographolide is comparable to paracetamol as a potentially weak pain killer, but it may present with reduced side-effects.

Type 2 nitric oxide synthase (iNOS or NOS2) is highly expressed in activated macrophages, which plays a crucial role in the pathogenesis of inflammation [46]. Induced by IL-6 and IL-1, aberrant NO production is directly involved in CRS pathogenesis and is known to cause vasodilation and hypotension, which are standard features of clinical CRS that require vasopressor administration [47]. In addition, both iNOS and PGE2 have been found to contribute to pain, swelling and cartilage destruction in inflammatory diseases such as those associated with the osteoarthritic joint [48]. As shown in Fig 1B, andrographolide inhibited NO in a dose-dependent manner and exhibited greater potency compared to the NSAIDs, with an $IC_{50}$ value of 7.4 μM (95% CI from 6.7 to 8.1 μM). The NSAIDs showed no significant NO inhibition until the concentration was increased to >100 μM. Diclofenac displayed the highest potency with an $IC_{50}$ value of 222 μM, which was still much higher than that of andrographolide. This result is in line with a previous study that demonstrated that andrographolide suppressed the expression of iNOS in macrophages and subsequently restored vasoconstriction in rat aortas treated with LPS [49]. To date, no NOS inhibitors are available for the treatment of inflammatory-induced pain. Andrographolide may serve as a potential therapeutic compound in the role of a NOS blocker, which could be beneficial for cardiac vasodilation.

TNF-α is an important proinflammatory cytokine that plays a central role in the cytokine storm. As seen in Fig 1C, andrographolide exhibited the most significant inhibition of TNF-α ($IC_{50}$ = 23.3 μM) compared to NSAIDs, which did not show any significant TNF-α inhibition at the tested concentrations. It should be noted that ibuprofen showed a weak TNF-α inhibition with an $IC_{50}$ estimated above 1500 μM.

As NSAIDs are not targeted at iNOS or TNF-α, it is not surprising that they show little to no activity against these two therapeutic targets. On the other hand, andrographolide displayed potent inhibition of both iNOS and TNF-α in addition to $PGE_2$, highlighting its broad anti-inflammatory activity *via* different mechanisms [50]. In addition, the potent inhibitory activity on TNF-α of andrographolide indicates its potential use against CRS. Therefore, further investigations were undertaken on the cytokine-inhibiting activity of andrographolide on THP-1 cells.

## Broad cytokine-inhibiting effects of andrographolide on LPS stimulated THP-1 cells

The cytokine-inhibiting activity of andrographolide in PMA-differentiated THP-1 cells under the stimulation of LPS was studied. Upon the stimulation of LPS, the cytokines TNF-α, IFN-γ, IL-1β, IL-2, IL-4, IL-6, G-CSF, GM-CSF and MCP-1 were upregulated significantly ($p < 0.05$), which were captured and quantified through the BioPlex 100 Bio-Plex system. Although several other cytokines were detected, they could not be quantified because their concentrations

**Table 2. The half maximal cytokines inhibitory concentrations of andrographolide, diclofenac, aspirin, paracetamol, and ibuprofen in LPS-induced THP-1 cells.**

| Cytokines | Andrographolide | | Diclofenac | | Aspirin | | Paracetamol | | Ibuprofen | |
|---|---|---|---|---|---|---|---|---|---|---|
| | $IC_{50}$ ($\mu M$) | 95% CI ($\mu M$) | $IC_{50}$ ($\mu M$) | 95% CI ($\mu M$) | $IC_{50}$ ($\mu M$) | 95% CI ($\mu M$) | $IC_{50}$ ($\mu M$) | 95% CI ($\mu M$) | $IC_{50}$ ($\mu M$) | 95% CI ($\mu M$) |
| TNF-α | 29.3 | 24.7 to 34.7 | ~469 | IF | >1000 | IA | >6000 | IA | ~1671 | IA |
| IFN-γ | ~31.4 | IF | ~162 | IF | >1000 | IA | >6000 | IA | ~1574 | IA |
| IL-1β | 18.1 | 5.1 to 63 | ~151 | IF | >1000 | IA | ~4362 | IA | ~1241 | IA |
| IL-2 | 35.7 | 28.3 to 45.0 | ~365 | IF | >1000 | IA | ~6000 | IA | ~1663 | IA |
| IL-4 | 32.8 | 28.2 to 38.3 | ~326 | IF | >1000 | IA | >6000 | 2017 to 20101 | ~1519 | IA |
| IL-6 | 12.2 | 9.1 to 16.2 | ~189 | IF | >1000 | IA | ~920 | IF | ~907 | IF |
| G-CSF | 31.90 | 22.49 to 45.26 | ~213 | IF | >1000 | IA | ~6831 | 3108 to 15012 | ~1800 | IA |
| GM-CSF | 65.2 | 31.5 to 135.0 | ~405 | IF | >1000 | IA | >6000 | IA | >1500 | IA |
| MCP-1 | 45.95 | 26.41 to 79.96 | ~314 | IF | >1000 | IA | ~2936 | IF | ~872 | IF |

(~)—Estimate IC50 due to poor curve fit.

IA—Insufficient activity to estimate an IC50 range.

IF—Insufficient curve fit to estimate an IC50 range (high hillslope).

* The data presented is the result of 2 independent experiments ($n = 2$).

lay outside the calibration range for the substances of interest, namely, IL-5, IL-7, IL-10, IL-12, and IL-13 (all below the detection limit) and IL-8 and MIP-1b (both higher than the highest standard).

As summarised in **Table 2**, andrographolide exhibited consistent inhibitory effects on multiple cytokines against LPS stimulation, with $IC_{50}$ values ranging from 12.2 to 65.2 μM. Most NSAIDs showed little to no activity ($IC_{50}$ values >150 μM). The most potent cytokine-inhibiting activity of andrographolide was seen in IL-6, with an $IC_{50}$ of 12.2 μM with a 95% CI range of 9.1 to 16.2 μM, indicating minimal variation. IL-6 is primarily considered a proinflammatory cytokine, contributing to host defence in response to infections and tissue injury, and its dysregulation is associated with the pathogenesis of chronic inflammation and autoimmunity [51]. Moreover, recent clinical trials have shown that IL-6 plays a central role in the mechanism of the cytokine storm, and serves as a predictor for disease severity and mortality in COVID-19 [52]. Thus, IL-6 and its receptor have been suggested as important therapeutic targets for cytokine storm, and tocilizumab, an IL-6 receptor (IL-6R) antagonist, was repositioned for the trials of cytokine storm against COVID-19 during the pandemic [53]. However, the cytokine-inhibiting activity of andrographolide is not restricted to IL-6 as it also influences other central cytokines.

TNF-α, IFN-γ and IL-1β contribute to the escalation of the cytokine storm through different modes of action. In influenza viral infection-induced cytokine storm, the reduction of TNF-α results in improved body weight and survival, despite a minimal impact on viral clearance, indicating that it may be a promising therapeutic target [54]. IFN-γ is a potent antiviral cytokine mediated through JAK-STAT pathway. However, its overexpression is linked to lung

injury [55]. The role of IL-1β in cytokine storm is associated with the induction of NLRP3 inflammasome, and its function is quite complex, promoting viral clearance and immune pathology. Our data indicated that andrographolide significantly inhibited TNF-α, IFN-γ and IL-1β simultaneously, with $IC_{50}$ values of 29.3, 31.4 and 18.1 μM, respectively. This result tentatively indicates andrographolide's potent activity in reducing cytokine release *via* different modes of action. In contrast, none of the tested NSAIDs achieved 50% inhibition within the tested concentration range (0–200 μM), and the $IC_{50}$ values were either estimated over 150 μM or out of the range at the upper calibration limit.

Andrographolide also demonstrated inhibitory activity in chemokines, including G-CSF, GM-CSF and MCP-1. Increased levels of these chemokines were also detected in COVID-19 patients who presented with acute respiratory distress syndrome [56]. The functions of these chemokines are associated with the recruitment of macrophages, neutrophils and other polymorphonuclear cells to inflammatory sites, which then escalates into an inflammatory cascade. Therefore, the suppressive effect of andrographolide could be beneficial in preventing inflammatory cascades. It should be noted that andrographolide also inhibited the release of IL-2 ($IC_{50}$ = 35.7 μM) and IL-4 ($IC_{50}$ = 32.8 μM), which play a vital role in the downregulation of immune responses [57]. IL-2 exerts both immunoregulatory and immunostimulatory activities, which are pivotal for cellular activation, and play an important role in primary T-cell responses and an essential role in secondary T-cell responses [57]. IL-4, a Th2-type cytokine, plays an active role in both the innate and adaptive immune response by suppressing Th1-type responses generated by the production of IFN-γ and TNF-α and inhibiting intracellular killing by macrophages [58]. Thus, the action of andrographolide in reducing IL-2 and IL-4 may not be desirable in the treatment of any cytokine storm and hyperinflammation associated with these cytokines, however, further *in vivo* studies are warranted to examine the overall effect of andrographolide in regulating the whole cascade of the cytokine storm.

## The inhibitory effect of andrographolide on LPS induced NFκB activation

Andrographolide is known to exert anti-inflammatory activity *via* the down-regulation of NFκB activation. While transfecting the cell line with multiple copies of NFκB to study other promoter elements of E-selection in ELAM9 can be interesting, it is beyond the scope of this study. The primary aim of this study was to directly compare the broad-based inhibition of plant-derived andrographolide and other NSAIDs against the cytokines involved in the NFκB inflammatory cascade, prior to any potential clinical application against the cytokine storm [59].

In this study, the activity of andrographolide was compared to that of NSAIDs using flow cytometry on ELAM9-RAW264.7 cells with NFκB green fluorescent protein (GFP). Upon lipopolysaccharide (LPS) stimulation, the ELAM9-RAW264.7 cells expressed NFκB GFP, which was then captured by the flow cytometer. As shown in **Table 3** and **Fig 2**,

**Table 3. The $IC_{50}$ values of andrographolide, diclofenac, aspirin, paracetamol and ibuprofen in inhibiting ELAM9-RAW264.7 cells with NFκB green fluorescent protein (GFP).**

| Assay | Andrographolide | | Diclofenac | | Aspirin | | Paracetamol | | Ibuprofen | |
|---|---|---|---|---|---|---|---|---|---|---|
| | $IC_{50}$ (μM) | 95% CI (μM) | $IC_{50}$ (μM) | 95% CI (μM) | $IC_{50}$ (μM) | 95% CI (μM) | $IC_{50}$ (μM) | 95% CI (μM) | $IC_{50}$ (μM) | 95% CI (μM) |
| NF-κB activation $IC_{50}$ | 26.0 | 23.4 to 29.0 | 508.3 | 400.3 to 645.4 | >1600 | - | >6000 | - | >1500 | - |

* The data presented is the result of 3 independent experiments (*n* = 3).

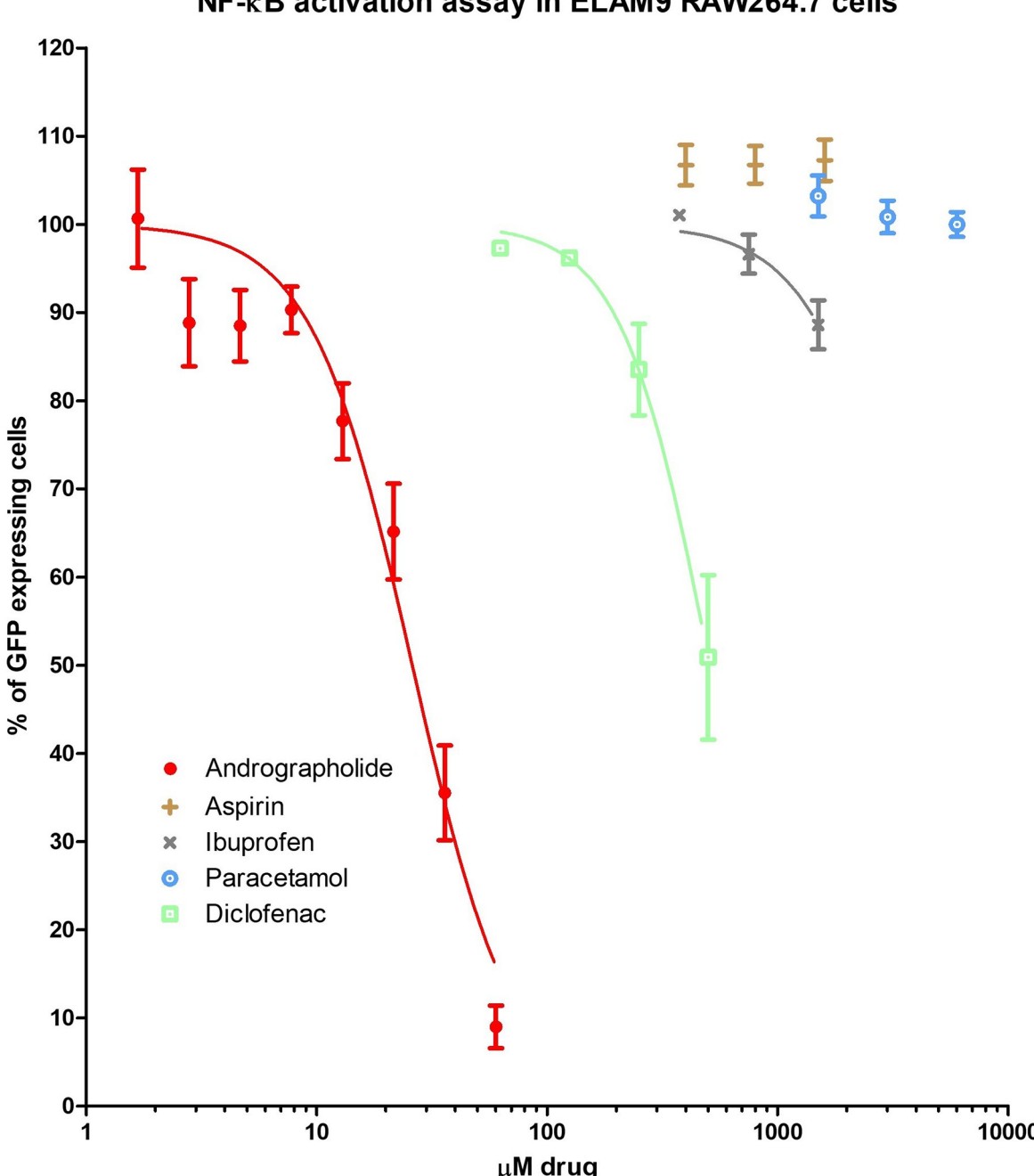

**Fig 2. Dose-response curves of andrographolide compared with ibuprofen, aspirin, diclofenac, and paracetamol in inhibiting NFκB (GFP) expressions on ELAM9-RAW264.7 cells measured by flow cytometry.** Error bars express standard error of the mean with $n = 3$ replicates.

andrographolide inhibited NFκB activation in a dose-dependent manner with $IC_{50}$ at 26.0 μM. Most of the tested NSAIDs did not impact NFκB, except for diclofenac which indicated an inhibitory trend ($IC_{50}$ at 508.3 μM).

Andrographolide was markedly more potent than the NSAIDs in downregulating NFκB activation. NFκB plays a central role in maintaining normal cell function and producing

inflammatory mediators in inflammatory conditions and acts as an upstream proinflammatory mediator. The NFκB-mediated signaling pathway interacts with cytokines, including IFNs, and cell survival. The NFκB pathway is often triggered by toll-like receptors upon exposure to viral pathogens, leading to host immune responses and the release of proinflammatory cytokines. It has been recently recognised that COVID-19 activates the NFκB pathway, like MERS and SARS-COV [60], leading to an increased level of inflammatory mediators. The inhibition of NFκB improved the survival of BALB/c mice and reduced SARS-COV-induced inflammation without influencing viral titers [61]. Thus, the potent and broad cytokine-inhibiting effect of andrographolide may be attributed to its down-regulation of NFκB activation.

In contrast, NSAIDs' defined mechanism of action simplifies the understanding and rationalisation of their anti-inflammatory effect. Their expressed side effects are easily linked to their mechanism of action because it is well understood. Andrographolide's widespread action makes its safety and any potential side effects harder to predict. However, *A. paniculata* has been used for thousands of years in Ayurvedic medicine and is considered safe [62–66]. Andrographolide has been shown to have a high therapeutic index, with a safety margin ($LD_{50}$) for administration intraperitoneally (11.46 g/kg), but phytochemical variation from plant-based extracts can affect the potential clinical efficacy [67,68]. Andrographolide has also been reported to exhibit gastro-protective and ulcer-preventive effects, which, in combination with its anti-inflammatory effects, could potentially make it a safer alternative to traditional NSAIDs [35].

Although this study highlights the promising potential of andrographolide compared to the widely used NSAIDs to address cytokine storm, it is limited, as only cell models were studied. The cytokine storm is a complex inflammatory event that is better represented in *in vitro* models and clinical settings. However, these results demonstrate andrographolide's clinical potential. The *in vitro* results also fail to compensate for the differing pharmacokinetic profiles of each drug, so direct comparison is complicated. It should also be noted that that there are several other plant-derived drugs such as resveratrol, tetrahydrocannabinol (THC) and cannabidiol (CBD) that have shown promising cytokine-inhibiting activity against the cytokine storm in recent studies when used as an adjuvant in COVID-19 treatment [69–72]. Future work could involve studies evaluating the efficacy of these drugs in comparison to andrographolide and andrographolide derivatives against the cytokine storm in a broader clinical context.

In summary, andrographolide is a promising candidate as an alternative to NSAIDs for broad inflammatory ailments and the management of the cytokine storm. However, further *in vivo* and human clinical studies are needed, along with a comprehensive multi-omics understanding of the current findings to confirm potential efficacy.

## Conclusion

Andrographolide was shown to possess potent inhibitory effects in LPS and IFN-γ induced PGE2, NO and TNF-α in the RAW264.7 cells and LPS-induced multiple cytokines in the THP-1 cells, including TNF-α, IFN-γ, IL-1β, IL-2, IL-4, IL-6, G-CSF, GM-CSF and MCP-1. The NO and cytokine-inhibitory properties of andrographolide were generally more potent than the common NSAIDs tested in this study (aspirin, ibuprofen, paracetamol and diclofenac). NSAIDs only exhibited higher potency in inhibiting $PGE_2$, where the activity of andrographolide was comparable to that of paracetamol. The broader cytokine-inhibiting activity of andrographolide was associated with the downregulation of the activation of NF-κB as tested in ELAM9 RAW264.7 cells. Overall, andrographolide may serve as a promising candidate therapeutic compound in regulating multiple cytokines and their associated inflammatory responses. Andrographolide has a broader anti-inflammatory effect than NSAIDs and may be

better suited to the complex nature of the inflammatory immune response. Further *in vivo* and human clinical studies are needed to confirm and verify the andrographolide's potential efficacy and side-effect profile.

## Supporting information

**S1 File. Supplement No.1 MTT data.**
(ZIP)

**S2 File. Supplement No.2 flow cytometry.**
(ZIP)

## Acknowledgments

We thank Mr. Dusko Pejnovic, Chief Executive Officer (CEO) of LIPA Pharmaceuticals Ltd., for his support in providing samples of *A. paniculata* extracts. The authors would like to dedicate this paper to the memory of the late Prof. Nikolaus J. Sucher (N.J.S.), who sadly passed away before this research could published. He made a significant contribution to the research presented in this paper. He will be missed.

## Author Contributions

**Conceptualization:** Mitchell Low, Cheang Khoo, Gerald Münch.

**Data curation:** Mitchell Low.

**Formal analysis:** Mitchell Low.

**Funding acquisition:** Gerald Münch.

**Investigation:** Mitchell Low.

**Methodology:** Mitchell Low, Cheang Khoo, Gerald Münch.

**Project administration:** Mitchell Low, Cheang Khoo, Gerald Münch, Chun Guang Li.

**Resources:** Cheang Khoo, Chun Guang Li.

**Software:** Mitchell Low, Muhammad A. Alsherbiny.

**Supervision:** Cheang Khoo, Gerald Münch, Chun Guang Li.

**Validation:** Mitchell Low.

**Visualization:** Mitchell Low, Harsha Suresh, Muhammad A. Alsherbiny.

**Writing – original draft:** Mitchell Low, Harsha Suresh.

**Writing – review & editing:** Mitchell Low, Harsha Suresh, Xian Zhou, Deep Jyoti Bhuyan, Muhammad A. Alsherbiny, Cheang Khoo, Gerald Münch, Chun Guang Li.

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
