## [Decision Letter · Decision Letter 0]

10 Apr 2024

PONE-D-24-06364The wide spectrum anti-inflammatory activity of andrographolide in comparison to NSAIDs: a promising therapeutic compound against the cytokine stormPLOS ONE

Dear Dr. Suresh,

Thank you for submitting your manuscript to PLOS ONE. After careful consideration, we feel that it has merit but does not fully meet PLOS ONE’s publication criteria as it currently stands. Therefore, we invite you to submit a revised version of the manuscript that addresses the points raised during the review process.

The manuscript by Low et al., titled "The Wide Spectrum Anti-inflammatory Activity of Andrographolide in Comparison to NSAIDs: A Promising Therapeutic Compound Against the Cytokine Storm," has been thoroughly reviewed. It presents significant findings on andrographolide's potential as an anti-inflammatory agent, offering insights into its utility against cytokine storms compared to NSAIDs. Both reviewers commend the manuscript's relevance to clinical, pharmaceutical, and medical biology research, predicting it will be well-received within the scientific community. However, for publication consideration, specific revision requested by reviewers are required to address issues of clarity, reproducibility, and accessibility. ==============================

We look forward to receiving your revised manuscript.

Kind regards,

Asif Ali

Academic Editor

PLOS ONE

2. Please amend lines 80 – 82 where you state “synthetic approaches to drug development have not lived up to their promise” to neutral language.

“This study was partially supported by a Research Partnership Grant (RPG) from the Western Sydney University and LIPA Phar-maceuticals (Prof. Nikolaus Sucher), and a Western Sydney University Research Grant Scheme (RGS) grant (Prof. Nikolaus J. Sucher and Prof. Gerald Münch).”

Reviewers' comments:

Reviewer's Responses to Questions

**Comments to the Author**

1. Is the manuscript technically sound, and do the data support the conclusions?

Reviewer #1: Yes

Reviewer #2: Yes

2. Has the statistical analysis been performed appropriately and rigorously? 

Reviewer #1: Yes

Reviewer #2: Yes

3. Have the authors made all data underlying the findings in their manuscript fully available?

Reviewer #1: Yes

Reviewer #2: Yes

4. Is the manuscript presented in an intelligible fashion and written in standard English?

Reviewer #1: Yes

Reviewer #2: Yes

5. Review Comments to the Author

Reviewer #1: The manuscript proposes addressing the gap between NSAIDs and cytokine storms by incorporating the natural product andrographolide. The study underscores the promising therapeutic potential of andrographolide as a broad-spectrum anti-inflammatory agent capable of effectively targeting cytokine storms, suggesting its viability as an alternative to NSAIDs in clinical management. Before finalizing the manuscript for publication, I would like to address a couple of points.

1. In the Materials and Methods section, please provide the catalog numbers of the reagents utilized to enhance accessibility for others.

2. Could you explain how THP-1 cell differentiation into macrophages was confirmed, including whether cell surface staining or morphological staining techniques were employed for validation? Any supporting figures should go to supplementary figures. Did you still maintain PMA after 24 differentiation or you used regular culture medium after differentiation? How was the cell viability during differentiation? Please explain this in a sentence.

3. How many cells were acquired and what flow rate was utilized for the flow cytometry analysis in the Materials and Methods section?

4. Please properly cite FlowJo v10 whether together with country of origin or company.

5. In Figure 1, please avoid using green and red colors together to prevent potential confusion for individuals with color blindness.

Reviewer #2: Review Comments

The manuscript "The wide spectrum anti-inflammatory activity of andrographolide in comparison to NSAIDs: a promising therapeutic compound against the cytokine storm." by Low et al. is a good research analysis of Andrographolide is a naturally derived bioactive compound. This research article provides sufficient evidence of new bioactive compounds with existing NSAIDs drugs. This manuscript is of interest to the clinical, pharmaceutical and medical biology researchers, as well as different biology researchers and I expect that the article will be well-cited. I have the following minor comments to consider.

1. Authors add details (like method, replicates, etc.) in their table’s legend.

2. Authors provide raw data of supplementary tables with details?

6. PLOS authors have the option to publish the peer review history of their article (what does this mean?). If published, this will include your full peer review and any attached files.

Reviewer #1: **Yes: **Sera Averbek

Reviewer #2: **Yes: **Pawan Kumar

---

## [Author Response · Author response to Decision Letter 0]

23 May 2024

Response to Editor, Journal Requirements:

Response: The manuscript has been amended to PLOS ONE standards. 

 2. Please amend lines 80 – 82 where you state “synthetic approaches to drug development have not lived up to their promise” to neutral language.

Response: Change made in manuscript (Ln 82-83). 

“This study was partially supported by a Research Partnership Grant (RPG) from the Western Sydney University and LIPA Phar-maceuticals (Prof. Nikolaus Sucher), and a Western Sydney University Research Grant Scheme (RGS) grant (Prof. Nikolaus J. Sucher and Prof. Gerald Münch).” Please state what role the funders took in the study. If the funders had no role, please state: "The funders had no role in study design, data collection and analysis, decision to publish, or preparation of the manuscript." If this statement is not correct you must amend it as needed. Please include this amended Role of Funder statement in your cover letter; we will change the online submission form on your behalf.

Response: Funding amendment added. Statement also included in the Cover Letter. 

Response: The citation list is complete and correct, including the inclusion of new citations.

Response to Reviewer #1: 

The manuscript proposes addressing the gap between NSAIDs and cytokine storms by incorporating the natural product andrographolide. The study underscores the promising therapeutic potential of andrographolide as a broad-spectrum anti-inflammatory agent capable of effectively targeting cytokine storms, suggesting its viability as an alternative to NSAIDs in clinical management. Before finalizing the manuscript for publication, I would like to address a couple of points.

1. In the Materials and Methods section, please provide the catalog numbers of the reagents utilized to enhance accessibility for others.

Response: The authors feel that adding catalog numbers for individual reagents would be unnecessary since catalog numbers for Sigma-Aldrich and other suppliers are easily accessible upon a cursory search. Further, some of the used reagents might be discontinued by suppliers at a future date, so catalog information might become obsolete. In any case, the purity for any used analytical standards are listed in the ‘Chemicals and Reagents’ section. 

2. Could you explain how THP-1 cell differentiation into macrophages was confirmed, including whether cell surface staining or morphological staining techniques were employed for validation? Any supporting figures should go to supplementary figures. Did you still maintain PMA after 24 differentiation or you used regular culture medium after differentiation? How was the cell viability during differentiation? Please explain this in a sentence.

Response: There was no staining used. THP-1 cells are non-adherent, when they differentiate, they adhere, so non adherent cells were washed away with fresh PMA free medium. Confluence was near 100% so no significant cytotoxicity was observed. Further clarification regarding MTT assay S1. supporting data has been added to the ‘Protocol for MTT viability determination’ section.

3. How many cells were acquired and what flow rate was utilized for the flow cytometry analysis in the Materials and Methods section?

Response: The number of cells seeded prior to flow cytometry is now clearly stated as 10,000 in the ‘Determination of multiple cytokines in LPS-stimulated THP-1 cells using a Bioplex cytokine assay’ section. Clarifications regarding the flow rate, flow cytometry parameters and S2. supporting data have also been added to the ‘The regulation of NF-kB signalling pathway using a FACS Canto II flow cytometer’ section.

4. Please properly cite FlowJo v10 whether together with country of origin or company.

Response: FlowJo v10 sourcing information added to the ‘The regulation of NF-kB signalling pathway using a FACS Canto II flow cytometer’ section.

5. In Figure 1, please avoid using green and red colors together to prevent potential confusion for individuals with color blindness.

Response: The authors consider the use of red and green to be unavoidable, since limited colour combinations are available for clear differentiation. Please note that all trace lines in Fig 1. have different legends for data points, unique for each therapeutic, with an inverted triangle used for Diclofenac (green) and a solid circle used for Andrographolide (red). 

Response to Reviewer #2: 

The manuscript "The wide spectrum anti-inflammatory activity of andrographolide in comparison to NSAIDs: a promising therapeutic compound against the cytokine storm." by Low et al. is a good research analysis of Andrographolide is a naturally derived bioactive compound. This research article provides sufficient evidence of new bioactive compounds with existing NSAIDs drugs. This manuscript is of interest to the clinical, pharmaceutical and medical biology researchers, as well as different biology researchers and I expect that the article will be well-cited. I have the following minor comments to consider.

1. Authors add details (like method, replicates, etc.) in their table’s legend.

Response: Replicate information has now been added to each table and figure. Replicates were n=3 for all data except Table 2 (THP-1 multicytokine test) where n=2. The reasoning for using n=2 replicates in the THP-1 cytokine inhibition tests were the high cost of the testing kits at the time of purchase (~ $5-10K AUD).

2. Authors provide raw data of supplementary tables with details?

Response: Information regarding supporting files S1. and S2. are now added in a separate section after the References in accordance with PLOS ONE formatting. Clarifications regarding the two supporting files have been added to the ‘Protocol for MTT viability determination’ and ‘The regulation of NF-kB signalling pathway using a FACS Canto II flow cytometer’ sections respectively.

---

## [Decision Letter · Decision Letter 1]

27 Jun 2024

The wide spectrum anti-inflammatory activity of andrographolide in comparison to NSAIDs: a promising therapeutic compound against the cytokine storm

PONE-D-24-06364R1

Dear Dr. Suresh,

We’re pleased to inform you that your manuscript has been judged scientifically suitable for publication and will be formally accepted for publication once it meets all outstanding technical requirements.

Kind regards,

Asif Ali

Academic Editor

PLOS ONE

Additional Editor Comments (optional):

Reviewers' comments:

Reviewer's Responses to Questions

**Comments to the Author**

1. If the authors have adequately addressed your comments raised in a previous round of review and you feel that this manuscript is now acceptable for publication, you may indicate that here to bypass the “Comments to the Author” section, enter your conflict of interest statement in the “Confidential to Editor” section, and submit your "Accept" recommendation.

Reviewer #1: All comments have been addressed

Reviewer #2: All comments have been addressed

2. Is the manuscript technically sound, and do the data support the conclusions?

Reviewer #1: Yes

Reviewer #2: Yes

3. Has the statistical analysis been performed appropriately and rigorously? 

Reviewer #1: Yes

Reviewer #2: Yes

4. Have the authors made all data underlying the findings in their manuscript fully available?

Reviewer #1: Yes

Reviewer #2: Yes

5. Is the manuscript presented in an intelligible fashion and written in standard English?

Reviewer #1: Yes

Reviewer #2: Yes

6. Review Comments to the Author

Reviewer #1: (No Response)

Reviewer #2: Accept this publication. Authors have answered all comments and there is no need of further revision.

7. PLOS authors have the option to publish the peer review history of their article (what does this mean?). If published, this will include your full peer review and any attached files.

Reviewer #1: **Yes: **Sera Averbek

Reviewer #2: **Yes: **Pawan Kumar

---

## [Editor Report · Acceptance letter]

8 Jul 2024

PONE-D-24-06364R1 

PLOS ONE

Dear Dr. Suresh, 

I'm pleased to inform you that your manuscript has been deemed suitable for publication in PLOS ONE. Congratulations! Your manuscript is now being handed over to our production team.

Kind regards, 

on behalf of

Dr. Asif Ali 

Academic Editor

PLOS ONE